

# Contribution of fluorescent primary biological aerosol particles to low-level Arctic cloud residuals

Gabriel Pereira Freitas[1,2], Ben Kopec[3], Kouji Adachi[4], Radovan Krejci[1,2], Dominic Heslin-Rees[1,2], Karl Espen Yttri[5], Alun Hubbard[6,7], Jeffrey M. Welker[3,8,9], and Paul Zieger[1,2]

[1]Department of Environmental Science, Stockholm University, Stockholm, Sweden
[2]Bolin Centre for Climate Research, Stockholm University, Stockholm, Sweden
[3]Ecology and Genetics Research Unit, University of Oulu, Oulu, Finland
[4]Department of Atmosphere, Ocean, and Earth System Modeling Research, Meteorological Research Institute, Tsukuba, Japan
[5]The Climate and Environmental Research Institute NILU, Kjeller, Norway
[6]IC3 - Centre for Ice, Cryosphere, Carbon and Climate, Institutt for Geovitenskap, UiT - The Arctic University of Norway, Tromsø, Norway
[7]Geography Research Unit, University of Oulu, Oulu, Finland.
[8]Department of Biological Sciences, University of Alaska Anchorage, Anchorage, USA
[9]University of the Arctic, Rovaniemi, Finland

**Correspondence:** Paul Zieger (paul.zieger@aces.su.se)

**Abstract.** Mixed-phase clouds (MPC) are key players in the Arctic climate system due to their role in modulating solar and terrestrial radiation. Such radiative interactions critically rely on the ice content of MPC which, in turn, also depend on the availability of ice nucleating particles (INP). INP sources and concentrations are poorly understood in the Arctic. Recently, INP active at high temperatures were associated with the presence of primary biological aerosol particles (PBAP). Here, we

investigated for a full year the abundance and variability of fluorescent PBAP (fPBAP) within cloud residuals, directly sampled by a multiparameter bioaerosol spectrometer coupled to a ground-based counterflow virtual impactor inlet at the Zeppelin Observatory (475 m asl), Ny-Ålesund, Svalbard. fPBAP concentrations ($10^{-3}$–$10^{-2}$L$^{-1}$) and contributions to coarse-mode aerosol (0.1 to 1 in every $10^3$ particles) within cloud residuals were found to be close to those expected for concentrations of high-temperature INP. Transmission electron microscopy also confirmed the presence of PBAP, most likely bacteria, within

the cloud residual samples. Seasonally, our results reveal an elevated presence of fPBAP within cloud residuals in summer. Parallel water vapor isotope measurements point towards a link between summer clouds and regionally sourced air masses. Low-level MPC were predominantly observed at the beginning and end of summer, and one explanation for their presence is the existence of high-temperature INP. In this study, we present observational evidence that fPBAP may play an important role in determining the phase of low-level Arctic clouds. These findings have potential implications for the future description of

sources of cloud condensation nuclei given ongoing changes in the hydrological and biogeochemical cycles that will influence the PBAP flux in and towards the Arctic.



## 1 Introduction

Mixed phase clouds (MPC) contain both cloud droplets and ice crystals (Korolev et al., 2003). Their interaction with solar and terrestrial radiation depends on their ice-to-droplet mixing ratio (Matus and L'Ecuyer, 2017). The phase of an MPC is

affected by the aerosol population in the cloud (Storelvmo, 2017), especially by the presence of particles that can facilitate the formation of ice within the cloud, the ice nucleating particles (INP; see e.g. Kanji et al., 2017). Therefore, a key element in an improved understanding of MPC in the Arctic is unraveling the sources, properties and concentrations of INP (Solomon et al., 2018).

The representation of MPC and other aerosol-cloud interactions are important sources of uncertainties in climate models,

impacting our ability to correctly estimate radiative forcing in the Earth's climate system (Szopa et al., 2021). This is especially true in remote regions, such as the Arctic, where measurements are scarce (Schmale et al., 2021) and low-level MPC are prevalent throughout the year (Kay et al., 2016; Morrison et al., 2012). The Arctic has experienced surface temperature increases that are two to four times higher than the global average (Rantanen et al., 2022), known as Arctic amplification (Wendisch et al., 2023). Clouds are believed to be one key contributor to the Arctic radiative budget, prompting the need to improve our

understanding of aerosol-cloud interactions in the Arctic (Schmale et al., 2021).

INP facilitate ice growth at temperatures above that of homogeneous nucleation (temperatures below -38 °C, Kanji et al., 2017). Complete or partial glaciation of a cloud radically changes its radiative properties and lifetime and can even trigger precipitation (Lensky and Rosenfeld, 2003; Stopelli et al., 2015). In the Arctic, high temperature INP have been observed on a seasonal basis (Porter et al., 2022; Sze et al., 2023) and have been linked to biogenic oceanic and terrestrial sources (Šantl-

Temkiv et al., 2019; Hartmann et al., 2020; Pereira Freitas et al., 2023) along with dust emissions (Tobo et al., 2019). Satellite observations show that the prevalence of MPC in the Arctic and Antarctic regions can be explained to a large degree by the presence of INP (Carlsen and David, 2022).

Primary biological aerosol particles (PBAP) are biological particles that are emitted directly from the source to the atmosphere. These can be, but are not limited to, microorganisms, biological functional parts, fungal spores or just fragments of

vegetation (Després et al., 2012; Fröhlich-Nowoisky et al., 2016). Some PBAP are efficient INP, even at high temperatures (>-15 °C, Tobo et al., 2013). This is due to their microphysical properties and/or their excretion of ice nucleating proteins (Pummer et al., 2015). In the Arctic, PBAP dominate the number of high-temperature INP in summer (Sze et al., 2023; Pereira Freitas et al., 2023).

Some PBAP, such as bacteria, have been observed in cloud water samples that showed cloud condensation (Bauer et al.,

2002, 2003) and ice nucleating abilities (Joly et al., 2013). These in-cloud bacteria undergo cloud processing (Khaled et al., 2021) and growth (Sattler et al., 2001). The offline methods used to sample bacteria, however, are limited in quantifying the abundance of PBAP (Huffman et al., 2020). To overcome such limitations, online methods can be used, such as those based on single-particle ultraviolet laser-induced fluorescence (Huffman et al., 2020), which have been shown to give reasonable estimates of PBAP concentration in real time (Freitas et al., 2022; Crawford et al., 2017, 2020). Given the close link between



PBAP and high-temperature INP (Šantl-Temkiv et al., 2019; Creamean et al., 2019; Pereira Freitas et al., 2023), obtaining PBAP concentrations inside cloud particles is one way to understand the impact of PBAP serving as INP in cloud glaciation.

The ground-based counterflow virtual impactor (CVI) has been successfully used in recent years to improve our process-level understanding of aerosol-cloud interactions in the Arctic, for example, by determining the size distributions (of sub-micrometer aerosol, Karlsson et al. (2021, 2022)), the chemical composition (Gramlich et al., 2023) or the black carbon concentration

(Zieger et al., 2023) of cloud residuals, i.e. particles which were involved in cloud formation or cloud processes. In this study, we present the first investigation of the contribution of PBAP to cloud residuals in the Arctic.

Some studies link INP or PBAP to local and regional sources (Pereira Freitas et al., 2023; Creamean et al., 2022) or transportation from lower latitudes (Meinander et al., 2022; Shi et al., 2022; Si et al., 2019) by using back trajectories as a tool to investigate air mass origin and to identify potential source areas. Another method to investigate the air origin is to use the

water isotope ratios (hydrogen, $\delta$D and oxygen, $\delta^{18}$O), which has been used to distinguish local and transported sources of air (Sodemann et al., 2008; Sjostrom and Welker, 2009; Bonne et al., 2015; Noone et al., 2011). This determination of source and transport history is possible because deuterium excess in water vapor and precipitation is largely controlled by the conditions at the point of evaporation (Merlivat and Jouzel, 1979). In the Arctic, water isotopic measurements have been used to distinguish between moisture sourced locally, in response to variations in sea ice coverage, and moisture sourced from distant locations

(Kopec et al., 2016; Bonne et al., 2019; Akers et al., 2020; Bailey et al., 2021). In Svalbard, the site of interest of this study, and among other locations, it has been shown that low deuterium excess values ($< 5‰$) are typically driven by air masses comprised of predominantly locally-sourced moisture while high deuterium excess values ($> 10‰$) are found in air masses comprised of predominantly distant-sourced moisture (Kopec et al., 2016). Pairing CVI measurements of cloud residuals with water vapor isotopic measurements can thus be used to better understand the origin of a given air mass and aid in the source

identification of INP and PBAP.

We investigate the presence and impact of PBAP in low-level Arctic clouds present at Zeppelin Observatory, Svalbard, and address the following research questions: (i) Can we identify PBAP within cloud particles of low-level Arctic clouds using an online single-particle instrument coupled to a CVI inlet? (ii) If so, to what extent are they present throughout the year and what are their respective sources? And finally, (iii) if present, can we identify an impact on the cloud phase?

## 75  2  Methods

### 2.1  Campaign description

The measurements were part of the Ny-Ålesund Aerosol Cloud Experiment (NASCENT) 2019-2020 campaign. A complete overview of the campaign is given by Pasquier et al. (2022). In short, for one year and a half, which coincided with the MOSAiC expedition in the central Arctic (Shupe et al., 2022), several state-of-the-art aerosol, cloud and meteorological measurements

from different platforms were taken concurrently at various locations around Ny-Ålesund in a combined effort to unravel the properties of clouds and aerosols in the Arctic. In this work, we focus on measurements taken at the Zeppelin Observatory located 475 meters above sea level close to the top of the Zeppelin mountain, 2 kilometers south of the village (Pasquier et al.,



2022). Due to the topography of the mountain, the wind tends to blow predominantly from the south or from the north, with very little influence from crosswinds (see e.g., Pasquier et al., 2022). The observatory was engulfed in low-level clouds for an extensive period of the campaign (approx. 34 %, when the visibility was below 1 km as measured by the visibility sensor, see next section). The entire setup is illustrated in Figure 1.

## 2.2 Cloud particle sampling

Cloud droplets and ice crystals were sampled using a ground-based counterflow virtual impactor (CVI) inlet (Brechtel Inc., USA, Model 1205). The CVI only samples larger particles (above approx. 6 $\mu$m in aerodynamic diameter) representing aerosol particles that have been activated into cloud droplets or ice crystals. It does so by applying a counterflow against the sample flow, which only larger particles have enough inertia to penetrate through. A more technical description of the CVI is given in Noone et al. (1988); Shingler et al. (2012) and a detailed characterization of the ground-based CVI present at the Zeppelin Observatory, together with the corrections that need to be applied, is given in Karlsson et al. (2021). In summary, the measured concentrations of the cloud residuals after correcting for an enrichment factor have to be multiplied by a factor of 2, to account for a mean droplet sampling efficiency of around 48 %. This factor was determined by comparing the coarse-mode particle concentration ($> 0.8$ $\mu$m) measured by the MBS during the CVI operation with the corresponding ambient coarse-mode particle concentration measured by a FIDAS 200S (Palas GmbH, Germany) located on the terrace of Zeppelin observatory (see Fig. S1 in the SI). This value compares remarkably well with the CVI sampling efficiency of 46 % determined by Karlsson et al. (2021) using the comparison of CVI observations with cloud microphysical measurements and the comparison of the aerosol size distribution of ambient and cloud residual accumulation mode measurements. After the cloud droplets and ice crystal penetrate through virtual impaction, they are dried in the counterflow air. Possible sampling artifacts, such as crystal shattering or particle capture by the wake effect, are discussed in detail in Karlsson et al. (2021). However, these artifacts that increase the number of particles are shown to be limited to particles in the Aitken mode (particles < 100 nm) and not the coarse mode, which is the focus size range of this work. In addition, coarse mode particles are commonly of primary origin; thus, it is likely that no major CVI sampling artifacts significantly influenced the results presented here.

A visibility sensor is coupled to the CVI inlet (Belfort Instrument, USA, Model 6400). Whenever the visibility falls below 1000 meters, indicating the presence of clouds according to the WMO (Spänkuch et al., 2022; WMO, 2008), the CVI inlet is meant to be turned on. For part of the observational period of this study, the CVI was turned on automatically. However, for certain periods due to severe icing conditions, it was turned on manually. Given the manual operation of the CVI inlet and fluctuation of visibility to values above 1 km (leading to a short automatic stop of the CVI inlet sampling) within a short period, several cloud events (CE) could be contained within a single cloud, and some clouds were not sampled at all. Despite our best efforts to obtain a balanced data set throughout all seasons, the issues with icing on the inlet during cold periods with supercooled liquid cloud droplets led to fewer samples in the winter months. It should be noted that the summer generally shows denser clouds with a higher cloud water content and lower visibilities during cloudy conditions (see Fig. S6 in Zieger et al., 2023). However, there are several CE successfully sampled during winter, thus covering all months of the year. The exception is April 2020, when the MBS did not function. An overview of the CE sampled is given in Table S1 (in the Supplementary



Information, SI). The first minute of every CE was discarded to remove possible contamination by particles remaining in the inlet from previous sampling and switching of the inlet.

## 2.3 Single-particle bioaerosol characterization

The single particle characterization of the cloud residuals is performed using a multiparameter bioaerosol spectrometer (MBS, University of Hertfordshire, U.K.). The MBS is a single-particle instrument based on ultraviolet-light-induced fluorescence. A more complete description of the instrument is given by Ruske et al. (2017). In summary, for our instrument, a laminar sample flow (0.315 lpm) shielded by a sheath flow (1.715 lpm, leading to a 2.03 lpm as a total flow) guides particles through the instrument. A continuous low-power laser scatters light off of particles, and their size is retrieved by the intensity of the
scattered light. The instrument can reliably measure the fluorescence of particles with an optical diameter of $0.8\,\mu$m or larger. Then, a xenon flashlamp is triggered shining at the particle with a 280 nm ultraviolet light. If the particle fluoresces, its emitted light is collected by two collection mirrors and focused onto a diffraction grating. The diffracted light is then focused onto a detector covering the visible range between the wavelengths of 300-615 nm over 8 equally distant channels. Following the previous work by Freitas et al. (2022), if the fluorescence is more than 9 times the fluorescence background and its main
signal sits at 364 nm (tryptophan emission channel, a common protein in microorganisms, see e.g. Pöhlker et al., 2012), the particle is classified as a fluorescent primary biological aerosol particle (fPBAP). A general drawback of using ultra-violet laser-induced fluorescence (UV-LIF) as the main classification method are uncertainties relating to over-counting (fluorescent particles erroneously classified as PBAP) and under-counting (potential non-fluorescent PBAP generally not being counted).

## 2.4 Water vapor isotope measurements

Continuous atmospheric water vapor isotopic measurements accompanied the CVI inlet sampling at the Zeppelin Observatory to assist in source identification of water vapor and air mass history. Water vapor concentration and isotopic ratios of oxygen ($\delta^{18}$O) and hydrogen ($\delta$D) were measured using a Picarro L2130-i isotope and gas concentration analyzer (Picarro, Inc., USA). Deuterium excess (d-excess or d) values were computed in the form of d $= \delta$D-8$\cdot \delta^{18}$O (Dansgaard, 2012). The Picarro analyzer was also located in the Zeppelin Observatory (Figure 1). Inlet tubing ($\approx 3$ m) sampled ambient air directly above the roof of
the laboratory building approximately 3 m from the CVI. Isotopic observations began on 14 November 2019 and continued through the end of December 2020 to overlap with most of the cloud observation window.

To calibrate the water vapor isotopic measurements, the Picarro analyzer is connected to a Picarro Standards Delivery Module (SDM). Data calibration and processing for the measurements at Zeppelin Observatory follow those made at Pallas, Finland, on a similar instrument (Bailey et al., 2021). Every $\approx 24$ hours, the SDM supplied two water samples of known isotopic
composition that bracketed the range of isotopic measurements to standardize the measurements. The two standards used were USGS45 ($\delta^{18}$O = -2.238‰, $\delta$D = -10.3‰) and USGS49 ($\delta^{18}$O = -50.55‰, $\delta$D = -394.7‰). These standards were used to correct for any offsets to the VSMOW-SLAP scale and assess any instrument drift during the measurement period, which was minimal for this period. Given the low water vapor concentration at times during this measurement period ($< 1000$ ppm), it is necessary to correct any instrument bias that might exist at these lower concentrations (Steen-Larsen et al., 2013). A



humidity experiment was carried out at the time of installation of the instrument and followed the protocol described by Akers et al. (2020) included the measurement of the two standard waters over a range of water vapor concentrations regulated by dry air. A humidity response curve was developed and applied to the dataset. Additional data quality control protocols followed the methods described by Bailey et al. (2021). Once quality control and calibrations were conducted, water vapor concentration and isotopic ratios ($\delta^{18}$O, $\delta$D, d-excess) were aggregated into 5-minute averages. The data were further aggregated to only times

when the CVI was sampling to appropriately pair the isotopic observations with a given CE. Given the instrument analytical error and error in the calibration process, we estimate uncertainty to be $< 0.3\,\permil$ for $\delta^{18}$O, $< 1.1\,\permil$ for $\delta$D, and $< 2.1\,\permil$ for d-excess. Error values are highest when water vapor concentration is lowest. However, for the purpose of this analysis, we only focused on times when clouds were present, which are related to times of relatively higher water vapor content, and thus the error in the isotopic measurements is generally lower than they would be across the entire dataset. Importantly, these

instrument- and analysis-based errors are significantly lower than the natural variability explored in this study.

## 2.5 Cloud type classification

Unlike in situ cloud residual sampling at the Zeppelin Observatory, the Cloudnet dataset was retrieved for the region around the village of Ny-Ålesund approx. 2 km away from the observatory (Nomokonova et al., 2019). Using a combination of remote sensing techniques, a vertically-resolved cloud classification of the air column is obtained (Illingworth et al., 2007). This

classification is explained in depth for Ny-Ålesund in Nomokonova et al. (2019, 2020). In short, at 20-meter intervals, the air column is classified according to its physical properties. This covers clear sky (CS), cloud droplets (CD), drizzle (DR), cloud droplets and drizzle (DR+CD), ice crystals (I), ice crystals and supercooled droplets (I+SCD), melting ice (MI), melting ice and cloud droplets (MI+CD), aerosol (A), insects (Ins) and aerosol and insects (A+Ins). Here, we focus on an altitude of 400 to 600 meters. Potential problems of this approach to cloud classification could include cases where the the cloud over Ny-Ålesund

might not be the same as that at the Zeppelin mountain or where a cloud is present at one site and not at the other. However, given the long sampling times (longer than 30 min), there is a good chance that the cloud would be present at both sites for at least a portion of the sampling time, which is sufficient for CE classification. Table S1 describes all CE and the availability of Cloudnet data.

For each CE, an ice-to-droplet ratio is derived using the Cloudnet data set. As previously done in Karlsson et al. (2021), this

value is calculated as the ratio of ice-related classification points (I, I+SCD, MI and MI+CD) to droplet-related classification points (CD, DR and CD + DR). For those CE with a mixed ratio between ice and liquid classifications ($>1\,\%$ and $<90\,\%$), a mixed-phase cloud (MPC) classification is given. The Cloudnet data was also used to obtain the height of the cloud top following the work by Chellini et al. (2022).

## 2.6 Auxiliary parameters

The ambient air temperature and relative humidity (at the Zeppelin Observatory) were measured by a weather station coupled to the CVI. Furthermore, the column air temperature was recovered from daily radio sounds taken in the village of Ny-Ålesund





(Maturilli, 2020) and using the HATPRO sensor located at the AWIPEV station (Rose et al., 2005). These air temperature curves were used to recover the daily and CE height in which the air temperature reached values lower than -15 °C.

For one CE in August 2020, a sample with coarse mode aerosol used for transmission electronic microscopy (TEM, for particles above 0.7 $\mu$m in aerodynamic diameter) was successfully sampled for 30 min at 1 lpm behind the CVI inlet and the particles were classified using the elemental composition described by Adachi et al. (2022). On the TEM grid, three PBAP were successfully identified (among the 133 particles analyzed and representing around ≈2% of the particles collected) based on their shape and composition, confirming the presence of PBAP in the cloud residuals.

## 2.7 Back trajectory analysis

Back trajectory ensembles were initialized at a height of 250 m at the latitude and longitude of the observatory, every hour for the days in which there were valid observations. The ensemble was generated by shifting the meteorological fields, whilst keeping the initialized starting point the same; in total 27 back trajectories were initialized in each ensemble. The length of the back trajectories was restricted to 5 days. Data points along each and every back trajectory (i.e. endpoints) were selected only if they resided within the mixed layer (as defined by the model/HYSPLIT output). The endpoints were temporally and spatially collocated with gridded sea ice daily data derived from satellite observations (Copernicus Climate Change Service (C3S) Climate Data Store (CDS), accessed on 12/07/2023) to ascertain the surface type directly below each endpoint, within the mixed-layer. All back trajectories were carried out using the Hybrid Single-Particle Lagrangian Integrated Trajectory model (HYSPLIT V5.2.1, Draxler et al., 1998; Stein et al., 2015), with the Global Data Assimilation System (GDAS) 1°x1° archive data being used for the metrological fields (https://www.ready.noaa.gov/data/archives/gdas1/, last access: 12/07/2023). The Python package PySPLIT (Cross, 2015) was used to generate the ensemble back trajectories.

## 3 Results and discussion

During the period from June 2019 to December 2020, the CVI sampled 209 CE that lasted at least 30 minutes. For each CE, the coarse-mode aerosol (optical diameter >0.8 $\mu$m) was characterized in a single-particle manner by the MBS, resolving the fPBAP contribution for each CE.

First, we present an overview and a detailed characterization of fPBAP found in cloud residuals (Section 3.1 and 3.2). Second, a case study of a mixed-phase cloud event (MPC, Sect. 3.3), a liquid phase cloud event (Sect. 3.4) and an ice cloud event (Sect. 3.5) will be presented and discussed. The last part includes an analysis of the annual cycle of all characterized CE (Sect. 3.6) and their source allocation (Sect. 3.7).

## 3.1 Characterization of fluorescent primary biological particles within cloud residuals

A summary of fPBAP found in cloud residuals throughout the campaign is shown in Table 1. Over 209 CE, representing a total of 812 hours, 527 fPBAP were detected by the MBS within cloud residuals. This accounts for less than 1 particle per cloud hour (or 18.9 liters sampled by the MBS). However, extremely low fPBAP concentrations are in the range of typical high-



temperature INP concentrations found in the Arctic ($10^{-4}$-$10^{-1}$, at activation temperature $\approx$-15°C, Creamean et al. (2022); Sze et al. (2023)). In summer, fPBAP concentrations ranged from $10^{-3}$–$10^{-2}\,L^{-1}$ (mean: $8.1\cdot10^{-3}\,L^{-1}$) and contributed up to 5% (mean: 0.03%) of the coarse mode particles. In winter, both the concentration and the relative contribution to the coarse aerosol particles were lower (mean: $4.2\cdot10^{-3}\,L^{-1}$ and 0.005%, respectively). Of all sampled CE in summer and winter, 67% and 45% contained at least one cloud residual fPBAP, respectively. Moreover, despite PBAP contributing significantly to the INP population in the Arctic (e.g., Pereira Freitas et al., 2023) they are not the only source (e.g., Tobo et al., 2019). Given the relatively high number of clouds containing fPBAP and the susceptibility of Arctic clouds to have their phase modulated by low concentrations of INP (Prenni et al., 2007), fPBAP could be relevant to aiding cloud glaciation processes in Arctic low-level clouds.

### 3.2 Transmission electron microscopy of coarse cloud residuals

For a CE on the 22nd of August 2020, one TEM grid with identified PBAP was successfully sampled for 30 minutes behind the CVI which overlapped with the MBS sampling. Using the elemental analysis described by Adachi et al. (2020, 2022), we assessed the probable nature of the aerosol in the coarse mode sampled on the grid. The TEM images (Figure 2-A,B,C) show 3 PBAP that were part of cloud residuals (out of the 133 particles analyzed from the TEM grid or around 2%). During the same cloud event, the MBS measured 5 fPBAP cloud residuals (Figure 2-D), accounting for 0.05% of the total coarse mode particles. Webcam images of the cloud are also shown in Figure 2-D. It should be noted that the cut sizes for the TEM grid and the MBS slightly differ (0.7 $\mu$m in aerodynamic and 0.8 $\mu$m in optical diameter, respectively). The deuterium excess for this CE was 0.3‰, which is a very low number, signaling that this cloud (water vapor) was probably locally sourced. Unfortunately, Cloudnet data was not available for this CE, so no assessment of the cloud phase could be made.

To the best of our knowledge, this is the first time PBAP were directly imaged in cloud residuals (as opposed to being collected in a cloud water sample, Bauer et al., 2002), directly indicating their possible role as cloud condensation nuclei and/or INP. However, it is difficult to draw further conclusions based on one sample, thus this result should be taken as a supporting analysis to the more comprehensive MBS analysis.

### 3.3 Case I: Mixed-phase cloud

For the second week of September, the Zeppelin Observatory was continuously in cloud from the 5th to the 8th of September 2020. At the beginning, the Cloudnet classification was mostly consistent with a liquid cloud, with an increase in ice contribution towards the second half. For the first CE the ice fraction was 12 %. The cloud glaciated and transitioned to containing almost only ice crystals before dissipating (up to 97% of ice, Figure 3-A). Visibility remained below 1000 meters for most of the cloud. Wind speeds hovered around 2 m/s with a persistent southerly direction except for a few hours when it switched to northerly winds (Figure 3-B). These long-lasting MPC are common for the Arctic (Morrison et al., 2012).

The d-excess from the cloud water vapor started at -5‰ and ended at around 0‰. This small change and the overall low values signal that the air mass was of local origin (Kopec et al., 2016), indicating that the cloud water vapor was regionally sourced (Figure 3-C). However, it should be kept in mind that the aerosol and water vapour source might not be identical. The





aerosol population could for example be a mix between regional (e.g., Pereira Freitas et al., 2023) and transported sources (e.g., Behrenfeldt et al., 2008; Geng et al., 2010). The coarse-mode aerosol comprises of larger particles that are effectively removed by dry and wet deposition (Stopelli et al., 2015), but can sometimes be transported to the Arctic from lower latitudes (Behrenfeldt et al., 2008). The precipitation process along the path of transported air masses will lead to the depletion of d-
250 excess and the wet removal of aerosol. Although we cannot use the d-excess to decisively link the cloud aerosol population to regional sources, the d-excess values and additionally performed trajectory calculations (Figure S2) point to a more pronounced contribution of regional sources of both vapor and aerosol.

The cloud temperature at 475 m above sea-level was 7.5 °C at the start of the cloud and continuously dropped to 0 °C towards the end of the cloud as it glaciated. For this day, the air temperature reached values of -15 °C only at approximately 4000 meters
above sea level and the height of the cloud top at approximately 1600 meters (temperature at cloud top height: ≈ -8 °C, Figure S3). Thus, a likely explanation for the presence of ice within this cloud is ice nucleation being started by high-temperature INP (Fan et al., 2017), of which fPBAP are part of (Tobo et al., 2013), or secondary ice formation due to, e.g. ice crystals being deposited by clouds higher up in the atmosphere (Lohmann et al., 2016).

As the cloud developed, fPBAP were clearly detected by the MBS within the cloud residuals. A total of 58 fPBAP were
260 found within the cloud (over 4 CE) accounting for 2 in every $10^4$ coarse mode particles. The presence of fPBAP could be one of the explanations for cloud glaciation at temperatures at which the presence of high temperature INP would be required. However, further studies assessing the role of other glaciation mechanisms are required to fully establish the impact of fPBAP (and INP) on low-level Arctic clouds. Nevertheless, a study by Carlsen and David (2022) suggests that MPC in the Arctic and Antarctic would only be feasible year-round, should such high-temperature INP be present.

## 265 3.4 Case II: Liquid cloud

This is a case study that includes five CE that took place between 12 and 17 July 2020. The Cloudnet classification of these CE generally remained between CD and DR (Figure 4-A), similar to that of a liquid cloud. Here, some of the CE, when visibility was low, did not completely coincide with the expected Cloudnet classification (e.g. Cloudnet classified the air above Ny-Ålesund as aerosol/clear sky, while the low visibility clearly indicated a presence of clouds at the observatory), exemplifying
the differences between what was measured at the peak of the Zeppelin mountain and that above the Ny-Ålesund village. This discrepancy is due to the differences in wind pattern between the two sites (Pasquier et al., 2022). This was one limitation of this study, as sophisticated cloud probes were not available at such a comprehensive time frame as the methods used here.

The cloud persisted over northerly winds and low to medium wind speeds (Figure 4-B). Deuterium excess values were quite low (0–5‰) and followed quite close the ambient air temperature that stayed always above 2.5 °C (Figure 4-C). During all 5
CE, the altitude at which the temperature reached - 15 °C hovered around 4800 meters and cloud top height was highly variable between 4 and 8 km (Figure S4). In fact, ice was present in the cloud at an altitude of 2000 meters. Given the high temperatures within the cloud at the Zeppelin Observatory, not even the presence of high-temperature INP would give rise to ice formation within it. An indication that despite the presence of INP, meteorological conditions were not favorable for ice formation at the lower levels of the cloud, but ice formation was seen higher in the column at temperatures still above - 15 °C. However, it is not



possible to clearly separate whether it is the result of primary ice nucleation or other glaciation mechanisms above that initiate secondary ice formation below.

Several fPBAP were sampled by the MBS during these CE (Figure 4-D). Removing the 4th CE of this case study, in which the MBS was not fully operational, the MBS sampled 99 fPBAP, representing approximately 0.01% (or 1 in every $10^4$ particles) of the total coarse mode. Throughout the summer, fPBAP are ubiquitous at the Zeppelin Observatory (Pereira Freitas et al., 285    2023) and the results presented here show directly that they are present within cloud residuals and possibly acted as cloud condensation nuclei. Under suitable conditions, they could contribute to glaciation and MPC formation.

### 3.5    Case III: Ice cloud

This is a case study of 3 consecutive CE that took place during October 28th and 29th, 2020. These CE all had completely glaciated clouds (Figure 5-A). Across these events, significant cooling occurred (from -2.5 to -10°C) as the winds shifted 290    to coming from the north and strengthened (10 ms$^{-1}$). Along with these changes, the d-excess shifted from relatively low values (0-5‰) to relatively high values (15-20‰), signaling a more transported source. During this cloud, the MBS detected 10 fPBAP (0.015% contribution to the coarse mode, Figure 5-D). The increase in deuterium excess during these events points to a shift to a more distant source of moisture in this winter case, and the temperature is low enough so that ice nucleation could have been started by INP of different activation temperatures along the cloud column. However, as mentioned above, the 295    moisture and aerosol sources can, but not necessarily, be the same; thus, the deuterium excess values are only indicative. Given the meteorological conditions (Figure S5), little can be said about the influence of PBAP on this cloud. However, this shows that fPBAP are as evenly present in clouds as they are in the atmosphere throughout the year.

### 3.6    Annual cycle of cloud parameters

For all 209 CE measured during the years 2019 and 2020, several parameters were averaged for each CE, such as ambient 300    temperature, deuterium excess and ice-to-droplet ratio; these were then subsequently grouped by month (Figure 6). Above panel A of Figure 6 the number of CE (# cloud events) is shown along with the total hours sampled (# hours sampled) for each month of the year. As can be seen, most CE were concentrated in summer, when sampling conditions were generally better and it is known that low-level clouds are also more present in the summer months (Illingworth et al., 2007; Taylor et al., 2019; Curry and Ebert, 1992; Maturilli and Ebell, 2018).

The concentration of coarse mode within CE was generally lower in summer and higher in winter months (Figure 6-A), due to the increased contribution of sea spray in the fall and winter months (Adachi et al., 2022; Zieger et al., 2010) at the Zeppelin observatory. This is the opposite of the contribution of fPBAP to the coarse mode, which was much higher in summer than in winter. The annual cycle of the cloud fPBAP population reflects that of the general fPBAP population at the Zeppelin Observatory (Pereira Freitas et al., 2023). These seem to reflect the expectation that fPBAP would act as efficient cloud nuclei 310    (Ariya et al., 2009). At the beginning and end of summer, when the contribution of fPBAP is higher and meteorological conditions are favorable, they potentially could contribute to the formation of MPC by acting as INP.



The deuterium excess within the CE (Figure 6-B) shows high values for winter, while showing low values for the remainder of the year. This implies that the moisture in winter air masses was mainly sourced from long-range transport (Kopec et al., 2016). This agrees well with reports on the influence of lower latitudes on the Arctic aerosol population in winter (Sharma et al., 2006). For the remainder of the year, a low d-excess implies that moisture, and possibly PBAP, were more regionally and/or locally sourced (Kopec et al., 2016; Delattre et al., 2015; Froehlich et al., 2002). These results point to a possible locally driven regulation of cloud formation around Svalbard, to a degree which we cannot estimate. The d-excess observations are further confirmed by back trajectory analyses linking lower values to long-range transport from terrestrial sources (Figure S2).

The ice-to-droplet ratio (Figures 6-C) shows that low-level clouds at Ny-Ålesund during the colder months (1-5 and 9-12) are mainly represented by ice clouds, and clouds during the month of July are mainly represented by liquid droplets. For June and August, most of the clouds included both phases and as such were most-likely MPC, which reflects the findings of Mioche et al. (2015). Figure 6-C shows that MPC were mostly present between 400 and 600 meters at Ny-Ålesund at the beginning and end of summer, when there are suitable meteorological conditions and a higher contribution of PBAP to coarse-mode aerosol.

The ambient temperature at 475 meters above sea level reached values as low as -25 °C in winter and values as high as 10 °C in summer (Figure 6-D). In the months that transition from summer to winter, such as May-June and August-September, the air temperature was on average around 0 °C. For the months of May through September, the altitude in which temperature reached -15 °C sat above 2500 meters, as can be seen in Figure 6-E) by both daily radio soundings and continuous HATPRO vertical temperature profiling above the village of Ny-Ålesund for each individual CE. The cloud top height of low-level clouds was much higher in the beginning of the year, reaching its minima in summer, where it stayed below 2500 meters. Thus, the temperatures across the cloud columns in summer point to facilitated ice formation in the presence of high-temperature INP.

These results are an early but clear indication of the contribution of fPBAP to MPC in the Arctic. In the future, a more comprehensive study that uses single-particle cloud particle probes (which also allows the differentiation between individual ice and droplet particles) at the Zeppelin observatory collocated with the CVI measurements and a detailed assessment of the cloud column is warranted to fully comprehend the degree in which fPBAP are acting as INP and thus play a role in Arctic MPC formation.

## 3.7 Potential sources of fluorescent primary biological aerosol particles within cloud residuals

A comparison of all clouds that had concurrent deuterium excess and Cloudnet data available is shown in Figure 7 by showing the deuterium excess rate vs. the fPBAP contribution to the coarse mode cloud residual concentration for the three different cloud regimes (as defined by the ice-to-droplet ratio) separately for the winter and summer season. In Figure 7-A, liquid clouds appear only in summer and present high ambient air temperatures, as expected (Ebell et al., 2020). Low d-excess values link the water vapor or moisture to air masses of predominantly local origin; these clouds also show a reasonable contribution of fPBAP. As previously noted by Gierens et al. (2020), local-scale aerosol sources could be important for cloud formation and evolution in the Arctic.

MPC are present mainly in summer and at mild temperature (-5 °C–5 °C, as shown in Figure 7-B). Most of them have a reasonable contribution of fPBAP to the coarse mode and originate from local air masses, as indicated by the low d-excess



values. These results agree with those of Gierens et al. (2020), where they estimate an increased contribution of local sources in summer.

Ice clouds are predominantly seen in winter, but some are seen in summer (Figure 7-C). Those in summer are present at mild temperatures (-5 °C to 5 °C) and low d-excess. Some of the summer ice clouds were highly enriched in fPBAP (values above 0.025%) with d-excess rates at around 10‰, indicating that some ice clouds could have formed through the presence of INP, of which fPBAP are a significant part. Furthermore, these clouds are strongly related to regional and/or local sources of air mass and aerosols, which were previously reported to be the main contributors to fPBAP in the Arctic (Pereira Freitas et al., 2023).

## 4   Conclusions

In this study, we used single particle fluorescence spectroscopy to identify fluorescent primary bioaerosol aerosol particles (fPBAP) within cloud residuals of low-level Arctic clouds. These cloud residuals were collected using a ground-based counter-flow virtual impactor inlet installed at the Zeppelin Observatory near Ny-Ålesund, Svalbard, over the span of a year and a half. fPBAP cloud residuals exhibited higher concentrations ($10^{-3}$–$10^{-2}$ L$^{-1}$) and a greater contribution (0.1 to 1 in $10^3$ particles) to the coarse-mode aerosol during the summer compared to winter ($10^{-4}$–$10^{-3}$ L$^{-1}$, and 1 in $10^4$–$10^5$ particles, respectively). Water vapor isotope data linked these fPBAP cloud residuals to regional sources. Despite their low abundance, the presence of fPBAP in the cloud residuals was confirmed by transmission electronic microscopy. Our analysis of meteorological data and cloud classification data using remote sensing, showed that the presence of fPBAP was associated with the prevalence of mixed phase clouds (MPC) at the beginning and end of summer. This suggests that MPC formation would be possible if high-temperature INP were present. Therefore, we present experimental and direct evidences that fPBAB, possibly from the local biosphere, could contribute to cloud formation in the Arctic. However, the degree to which they influence cloud glaciation and MPC formation in general would require further investigation both experimental (e.g. better quantitative assessment of the cloud phase) but also using modelling approaches.

*Data availability.* The data will be available at the Bolin Centre for Climate Research database (DOI will be added later).

*Author contributions.* GF, RK and PZ performed MBS and GCVI measurements. KA performed TEM measurements. GF analyzed MBS, GCVI and auxiliary data and wrote manuscript with contributions from all co-authors. BK, AH and JW performed water isotope measurements and performed analysis. DH-R performed back trajectory analysis. KEY provided valuable insights to the discussion and writing of the manuscript. PZ conceived the study. All authors read and commented on the final version of the manuscript.

*Competing interests.* At least one of the (co-)authors is a member of the editorial board of Atmospheric Chemistry and Physics.



*Acknowledgements.* The authors would like to acknowledge the Norwegian Polar Institute for their long-term support at Zeppelin Observations. This research was supported by the Swedish Research Council (grant no. 2018-05045), the Knut och Alice Wallenbergs Stiftelse
(ACAS project grant no. 2016.0024) and the the Swedish Environmental Agency (Naturvårdsverket). This project has received funding from the European Union's Horizon 2020 research and innovation programme under Grant Agreement 821205 (FORCeS). This project has received funding from the European Union's Horizon 2020 research and innovation programme under grant agreement no. 101003826 via project CRiceS (Climate Relevant interactions and feedbacks: the key role of sea ice and Snow in the polar and global climate system). We thank the Environmental Research and Technology Development Fund (JPMEERF20232001) of the Environmental Restoration and Con-
servation Agency of Japan and the Arctic Challenge for Sustainability II (ArCS II) (JPMXD1420318865). Water isotopic measurements at Zeppelin were supported by the Academy of Finland award to Welker and Hubbard. The authors thank Valtteri Hyoky for on-site support and data processing for the isotopic measurements at Zeppelin. The authors recognize NPI for its long-term support of measurements at the Zeppelin Observatory.



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





**Table 1. Summary of detected fluorescent primary biological aerosol particles (fPBAP) inside low-level Arctic clouds.** Summer include the months June to September, while winter refers to October to May.

| | fPBAP measured (#) | Number of cloud events (#) | Total hours of sampled clouds (H) | Clouds containing fPBAP (%) | fPBAP conc. (mean, $10^{-3}$L$^{-1}$) | fPBAP conc. (median, $10^{-3}$L$^{-1}$) | fPBAP contr. to coarse mode (mean, %) | fPBAP contr. to coarse mode (median, %) |
|---|---|---|---|---|---|---|---|---|
| Summer | 476 | 156 | 612 | 67 | 8.1 | 4.8 | 0.032 | 0.012 |
| Winter | 51 | 53 | 200 | 45 | 4.3 | 0 | 0.005 | 0 |
| $\frac{\text{Summer}}{\text{Winter}}$ | 9 | 2.9 | 3.1 | 1.47 | - | - | - | - |



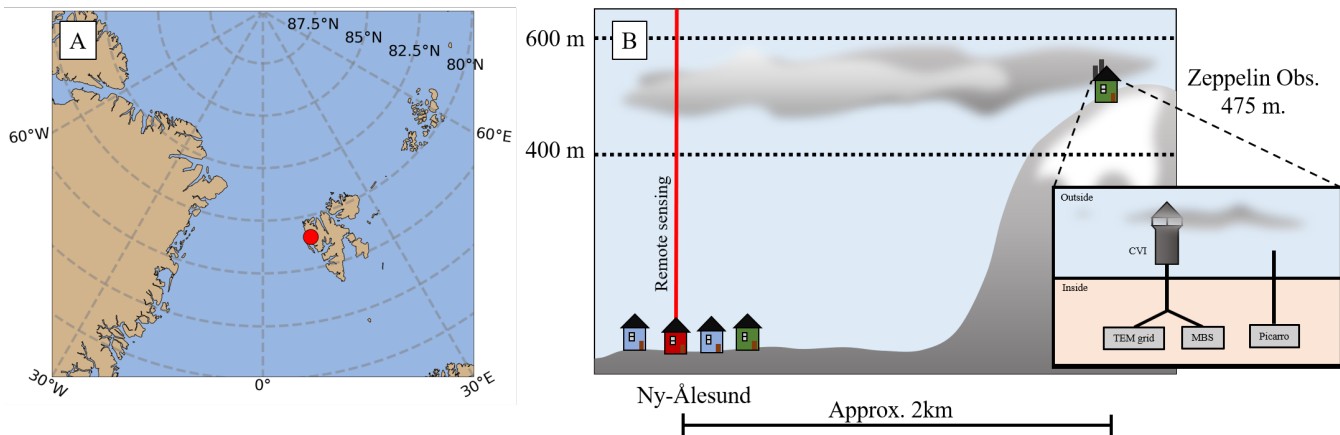

**Figure 1. Sampling location and measurement setup.** a) Location of Ny-Ålesund on the Norwegian archipelago of Svalbard (red dot). b) Schematic demonstrating the positioning of the different measurements in the town of Ny-Ålesund, where remote sensing took place, and at the Zeppelin Observatory (475 m asl), where in-situ cloud, aerosol and water vapour measurements were performed. For the cloud characterization via Cloudnet, the altitude between 400 and 600 meters was taken into account (dashed line). At the Zeppelin Observatory the TEM grid sampler and the MBS sampled cloud residuals from a CVI inlet. The Picarro sampled water vapour from its own gas-phase inlet.



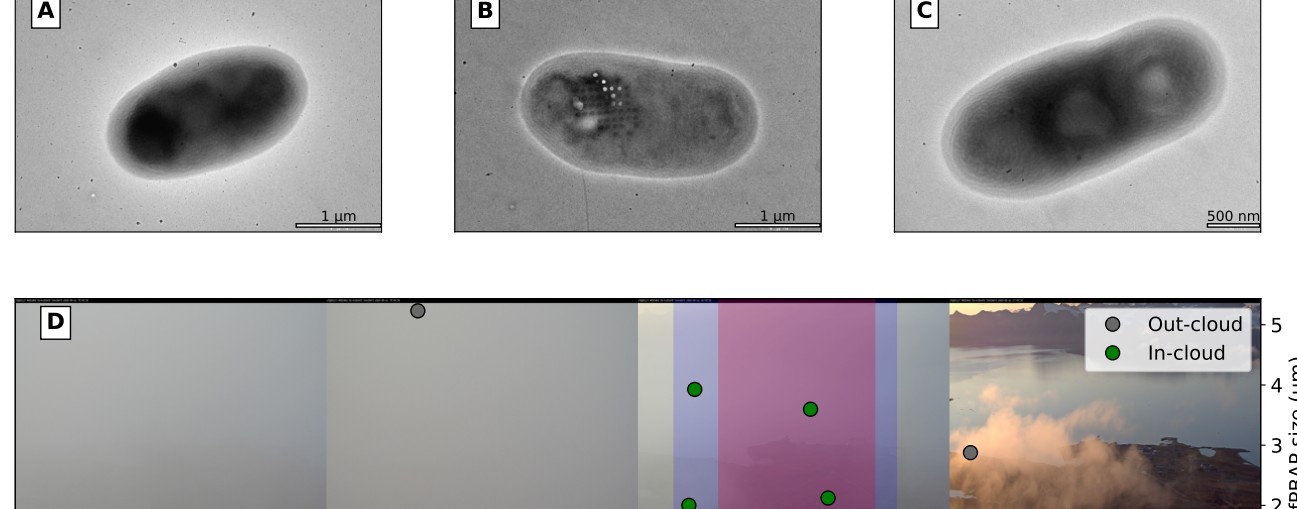

**Figure 2. Example of PBAP within cloud residuals as identified via TEM and MBS analysis.** Panel a-c show identified PBAP particles within the TEM sample taken on the 22nd of August 2020 (sampled behind the CVI inlet). Panel d shows webcam images (taken from Pedersen, 2013) of the cloud event and the periods of the CVI (blue area) and TEM sampling (pink area) period. In addition, the fPBAP particles identified by the MBS are shown in the background as a function of their size (right axis). Out-of-cloud measured fPBAP (MBS sampling from the whole-air-inlet) are shown for context. Dots on the particle shown in panel B are due to beam damage on the particle by the electron beam.

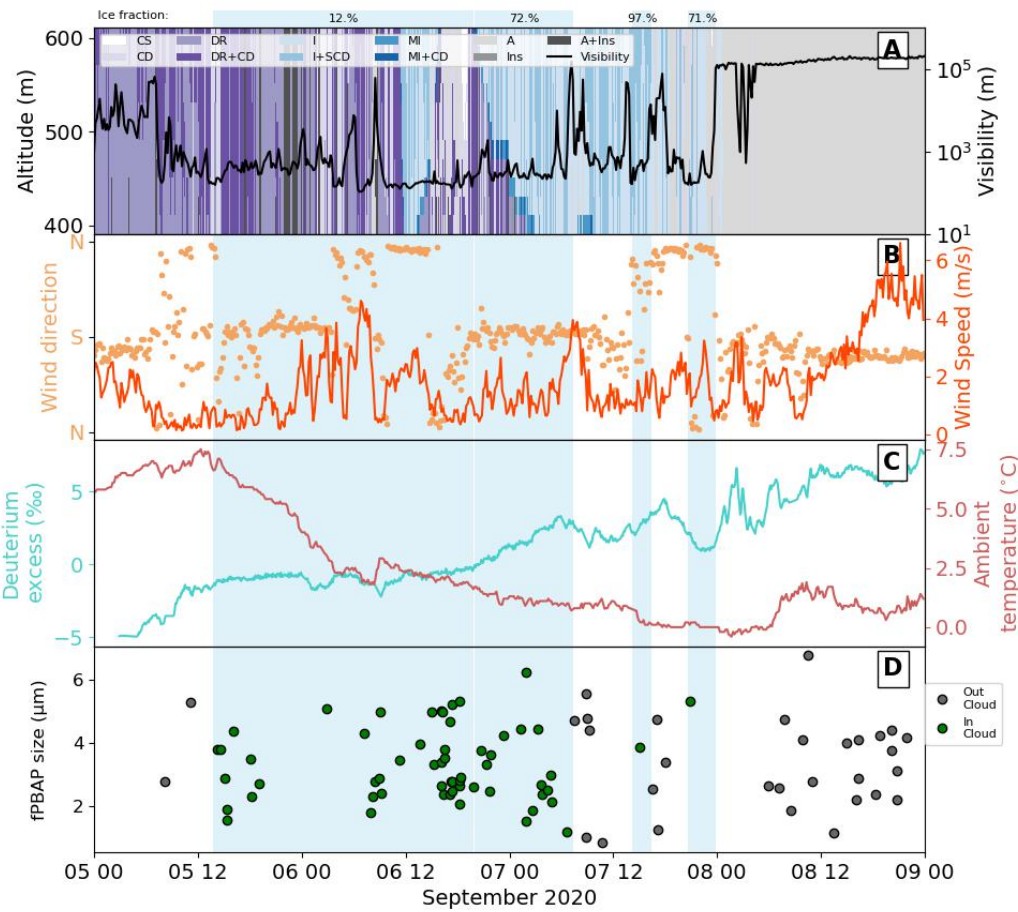

**Figure 3. Example of a mixed-phase cloud event measured in September 2020.** a) Cloudnet classification (at the height of Zeppelin observatory) and visibility (directly measured at Zeppelin observatory). Light blue boxes above and in the panels below indicate periods where CVI sampling occurred along with the ice-to-droplet ratio of each CVI cloud event calculated using the Cloudnet data. b) Wind direction and speed at the Zeppelin observatory. c) Water vapor deuterium excess and ambient temperature. d) Fluorescent primary biological aerosol particles (fPBAP) measured by the multiparameter bioaerosol spectrometer (MBS) categorized by size. Out-cloud PBAP are shown for context. Observations from panel b-d were taken at the Zeppelin observatory.



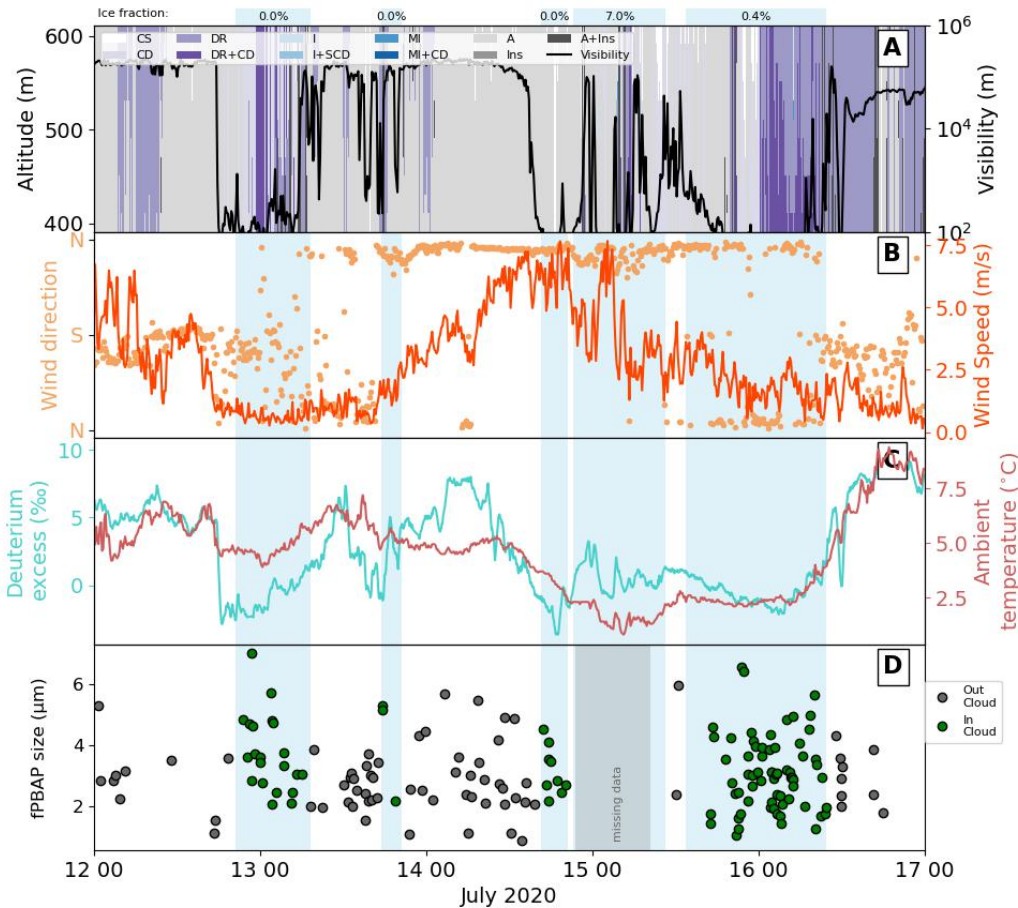

**Figure 4. Example of a droplet cloud event measured in July 2020.** a) Cloudnet classification (at the height of Zeppelin observatory) and visibility (directly measured at Zeppelin observatory). Light blue boxes above and in the panels below indicate periods where CVI sampling occurred along with the ice-to-droplet ratio of each CVI cloud event calculated using the Cloudnet data. b) Wind direction and speed at the Zeppelin observatory. c) Water vapor deuterium excess and ambient temperature. d) Fluorescent primary biological aerosol particles (fPBAP) measured by the multiparameter bioaerosol spectrometer (MBS) categorized by size. Out-cloud fPBAP are shown for context. Gray shaded area represents missing data from the MBS. Observations from panel b-d were taken at the Zeppelin observatory.



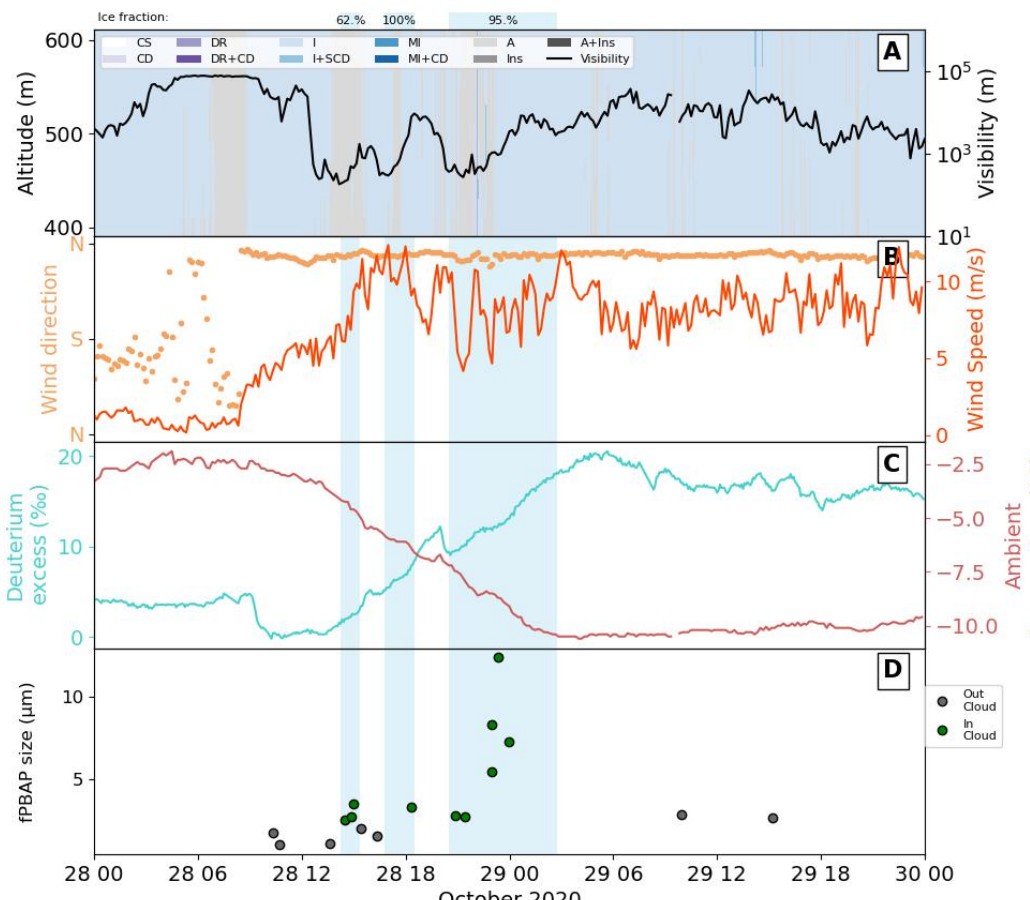

**Figure 5. Example of an ice cloud event measured in October 2020.** a) Cloudnet classification (at the height of Zeppelin observatory) and visibility (directly measured at Zeppelin observatory). Light blue boxes above and in the panels below indicate periods where CVI sampling occurred along with the ice-to-droplet ratio of each CVI cloud event calculated using the Cloudnet data. b) Wind direction and speed at the Zeppelin observatory. c) Water vapor deuterium excess and ambient temperature. d) Fluorescent primary biological aerosol particles (fPBAP) measured by the multiparameter bioaerosol spectrometer (MBS) categorized by size. Out-cloud fPBAP are shown for context. Observations from panel b-d were taken at the Zeppelin observatory.



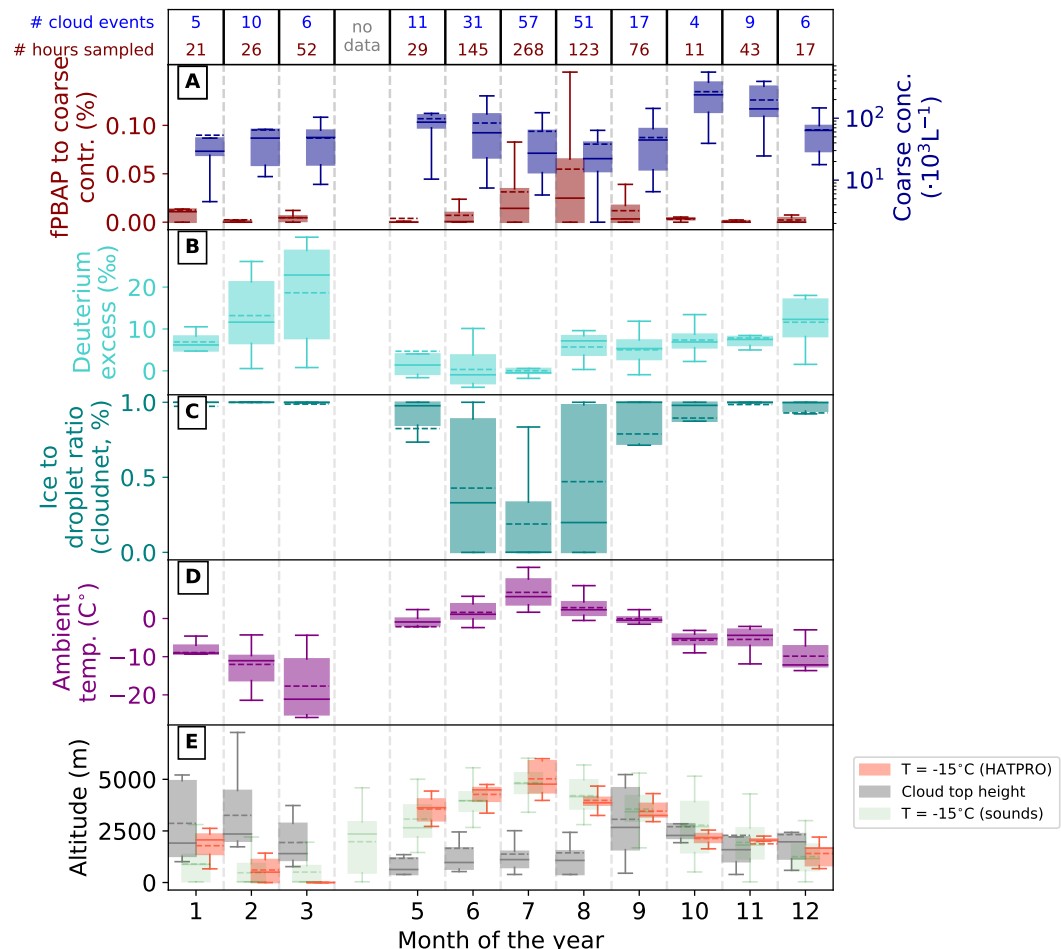

**Figure 6. Annual cycles of all relevant bioaerosol, water vapor, cloud and meteorological parameters during cloud events.** Above panel A, the number of cloud events and sampled hours per month of the year. For a detailed availability of data please see Table S1. These values refer to all datasets except for temperature soundings at panel e. a) Coarse aerosol (D $> 0.8\,\mu$m) concentration as measured by the multiparameter bioaerosol spectrometer, along with the contribution of fPBAP to the coarse mode (at Zeppelin Observatory). b) Water deuterium excess (at Zeppelin Observatory). c) Ice-to-droplet ratio, as calculated using Cloudnet data (measured above Ny-Ålesund at the height of Zeppelin Observatory). d) Ambient temperature (at Zeppelin Observatory). e) Altitude when air temperature is -15° as measured by daily atmospheric sounds from 2019-06 to 2020-12 and by the HATPRO measured above Ny-Ålesund at the height of Zeppelin Observatory. Furthermore, the cloud top height is also shown, as derived using Cloudnet data.



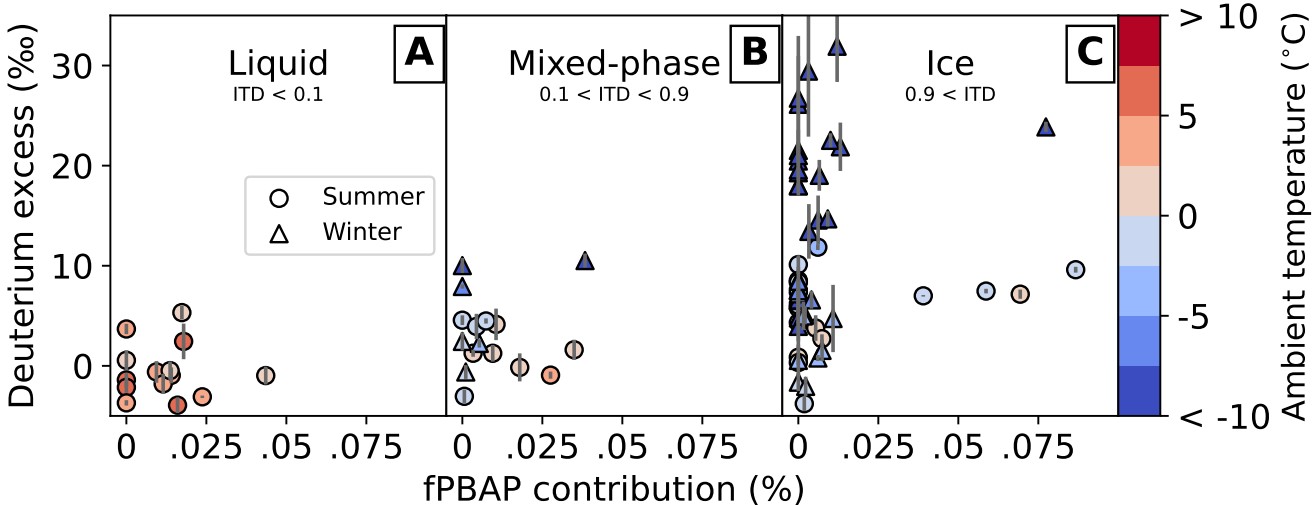

**Figure 7. Deuterium excess vs. fPBAP contribution to aerosol coarse mode number concentration.** Relationship shown for a) liquid, b) mixed-phase and c) ice cloud events as classified by the ice-to-droplet (ITD) ratio. Color-code represents the mean ambient temperature at 500 meters a.s.l. Triangles denote clouds in winter while circles show summer cloud events. Individual points are arithmetic mean values and error bars represent the corresponding standard deviation of the deuterium excess.