# Peer review of "Supporting Information: Contribution of fluorescent primary biological aerosol particles to low-level Arctic cloud residuals"

_EGUsphere, 2023_

## Author Comment (AC1)

Dear reviewers,

We first would like to thank you for your time and your helpful comments. We agree that the our original text was to some extent lengthier than necessary and that some of the results and conclusions were not perfectly aligned. To improve the clarity of our manuscript, we have moved two of the case studies (the cases about the pure ice and droplet clouds) to the supplement and moved the mixed-phase cloud case study after presenting the overall results. In addition, we thoroughly revised and improved language and structure. We hope that the made changes appropriately addressed your concerns. Our reply to your individual comments follow in blue color and changes in the text are given in red color.

**1 Reviewer 1**

This paper provides an overview of efforts to collect information on fluorescent primary biological aerosol particles on low-level Arctic clouds, using observations collected on Svalbard over two years. In general, the paper provides new perspectives on an important and timely topic, and offers new information to the community. While I was initially very excited to read the article and learn about the results, in the end I found the paper to be a bit lacking. There is a lot of focus on methods, an overview of some (very limited and not very well connected) case studies, and then some information on broader statistics. However, after reading all of this, the primary (only?) takeaway that I had was that fPBAPs exist and are present in the vicinity of clouds. Clearly there should have been more to it than that? If so, I don't believe that the authors did a great job of highlighting their outcomes and main take-home messages.

Given all of this, I recommend that this paper be considered for publication after minor revisions. I say minor not because the recommended revisions are easy, but because they don't require significant additional analysis and/or a total re-structuring of the manuscript. I really do think that this work represents a meaningful contribution and that the results should be shared with the community. However, I believe that the paper should be carefully refined to ensure that the results are clearly communicated and that the primary messages are highlighted and come with some broader discussion and context. Beyond this, please see additional comments below.

We thank the reviewer much for their helpful and constructive comments. We understand that the text was in parts lengthier than necessary along with too many repetitions. We also understand that two of the case events (liquid and ice clouds) were more examples than actual conclusion-drawing results. Thus we have decided to move them to the supplement but still shortly discuss them in the text. Along with these changes we have also restructured and improved the result and discussion section for better clarity. Finally, as was also pointed out by the other reviewer, the conclusion section was revised to better reflect the overall results of our work.

**1.1 Major Comments:**

- Case III (Ice Cloud): I didn't get much useful information out of this case study. It almost feels as though it was tacked on just to say that an ice cloud was looked at. As a general comment it would have been good to include some language that highlights the unique elements of each of these cases and perhaps contrasts those elements with the other cases. From all three, I got the message that fPBAP are present – what else should I have learned?

  Indeed, we agree with the reviewer in that there is not much to learn from the ice cloud case study aside from being an example. After further thoughts, we also think that not much new is learned from the liquid cloud case (except that indeed fPBAP are present in cloud residuals in both cases). For that reason we have moved these sections to the supplement but continue to shortly reference them in the text. As mentioned above, we now show the overall results first and then show the mixed-phase cloud example in the main manuscript (since this is the more relevant case). Following the reviewers comments, we believe that this change cleared the manuscript and allowed us to better discuss and clarify the main results.

- Given the potential for the ocean surface to be a significant source of PBAP and fPBAP, it seems as though the paper could use some additional discussion on the general state of regional sea ice. It is a more complicated scene than in many other parts of the Arctic due to the highly dynamic region that these samples are collected in.

  We completely agree with the reviewer. In our previous work at Zeppelin Pereira Freitas et al. (2023), we have found evidences that the majority of the fPBAP at the Zeppelin Observatory were of terrestrial origin. Nonetheless, we couldn't estimate the degree of contribution of marine PBAP to the overall population. We believe that this could only possibly be done by applying DNA-based techniques as trajectory based analysis

are limited to properly estimate the contribution of local sources. We have added a few sentences to the discussion section to better express that the sea ice and ocean are contributing to the PBAP, but to an extent which we cannot estimate:

"These results point to a possible regionally driven regulation of cloud formation around Svalbard, to a degree which we cannot estimate. The d-excess observations combined with back trajectory analyses link lower values to transport from terrestrial sources (Figure S6). These results point to a more pronounced terrestrial origin of cloud residual fPBAP, corroborating previous work where fPBAP detected at Zeppelin Observatory were found to originate to a large extend from regional and land-based sources (Pereira Freitas et al., 2023), including the polar semideserts that dominate the tundra of Svalbard (Welker et al., 1993; Wookey et al., 1995). Nonetheless, the ocean and sea ice can be a significant source of fPBAP and INP (Creamean et al., 2019). Moreover, the Svalbard region is notorious for being strongly affected by the Arctic amplification, prompting a dramatic change in annual sea-ice coverage (Urbański and Litwicka, 2022). Thus, although our results point to terrestrial sources, fPBAP sourced from regional marine and ice sources can still significantly contribute to the fPBAP population to a degree that is hard to estimate using our methods. DNA based techniques, such as those applied by Šantl-Temkiv et al. (2019) could better constrain the sources of fPBAP in the Arctic."

- In general, I found section 3.6 (Annual cycle of cloud parameters) sloppy and rambling. This section seems like it should be one of the most important ones in the text, yet there are no clear messages. Please take time to rework and expand on this section to ensure that you clearly communicate the most important points.

  We agree with the reviewer and the section has been reworked to highlight the most important aspects of this section along with placing it further up in the text. The new section reads as:

[revised manuscript text omitted]

- Section 3.7: This section feels like an afterthought. Should there be relationships between deuterium excess and fPBAP concentrations? What is a "reasonable contribution" (e.g., line 345)? Please be more quantitative and help the reader understand what might be expected and why. To me it doesn't necessarily make sense to segregate this by liquid, mixed-phase and ice clouds. Why? Are there differences expected? Or are those just seasonal? More detail and information is required in this section.

We agree with the reviewer and have reworked the section (including its title) to be more clear on our results from this analysis. We have also placed this section further up in the text. The new text follows:

"**Relationship between cloud phase, bioaerosol contribution and isotope ratio**

Figure 4 shows the d-excess rate vs. the fPBAP contribution to the coarse-mode cloud residual concentration for three cloud regimes grouped by winter and summer seasons. In Figure 4-A, liquid clouds appear only in summer and present high ambient air temperatures, as expected (Ebell et al., 2020). Low d-excess values link the water vapor to air masses of predominantly regional origin. These clouds had a fPBAP contribution of 0.01%. MPC are present mainly in summer and at mild temperature (from -5 °C to 5 °C, Figure 4-B). Most of them seem to originate from regional air masses, as indicated by the low d-excess values. For MPC at mild temperatures (from -5°C to 5°C), fPBAP contribution was 0.01% or higher. Ice clouds were predominantly seen in winter (Figure 4-C). Those observed in summer were present at mild temperatures (-5 °C to 5 °C) and typically had a low d-excess value. A few ice clouds were highly enriched in fPBAP (values above 0.025%) with d-excess rates at around 10‰, indicating a mix between regional and transported sources.

These results show that summer clouds containing ice were present at mild temperatures, often containing fPBAP (>0.01%), indicating the role of fPBAP in ice formation. For all cloud phase regimes, fPBAP were mostly present at lower d-excess values. Regional aerosol sources are important for cloud formation and evolution in the Arctic (Gierens et al., 2020), and this seems to reflected here."

- The "Conclusions" section reads more like a summary, essentially just repeating the findings laid out in the rest of the paper. I would recommend that the authors consider highlighting these points within the actual text, and then use section 4 to offer more discussion and put these observations into broader context.

We agree with the reviewer that the conclusions were not properly laid out and have completely overhauled the section. This comment has also been made by the other reviewer, so we have paid special attention in reworking the conclusions. The new conclusions read as follows:

"Within this work, we showed that fluorescent primary biological aerosol particles (fPBAP) are found within cloud residuals and possibly contributed to the formation of low-level Arctic clouds. This was achieved, for the first time, by direct observations using a ground-based counterflow virtual impactor inlet combined with online and offline particle sampling techniques. This approach avoided indirect proof of the relevance of fPBAP on cloud properties, for example, when using correlations of INP with fPBAP concentrations as done previously (Pereira Freitas et al., 2023). fPBAP exhibited higher concentrations ($10^{-3}$–$10^{-2}$ $L^{-1}$) and contributions (0.1 to 1 in $10^3$ particles) to the coarse-mode cloud residuals in summer compared to winter ($10^{-4}$–$10^{-3}$ $L^{-1}$, and 1 in $10^4$–$10^5$ particles, respectively). In summer, water vapor isotope data linked clouds

to regional sources. Thus, fPBAP most likely originated from the biosphere around Svalbard. The presence of fPBAP was associated with the prevalence of mixed phase clouds at the beginning and end of summer. Here, we present experimental and direct evidences that fPBAP contribute to ice formation in Arctic low-level clouds. However, cloud formation is a complex phenomena involving meteorology as well as interlinked cloud and aerosol microphysical and chemical processes. Thus, the degree to which fPBAP influence cloud glaciation in general would require further investigation both experimental (e.g. by quantitative assessment of the cloud phase using single particle cloud probes) but also using modelling approaches. Future work should also include filter sampling for genetic analysis to identify the biological material and origin, in addition to parallel sampling of bioaerosols within cloud residuals and interstitial aerosol to assess whether certain microorganisms are more likely to act as cloud condensation nuclei."

Minor Comments:

- Line 2: This is true, but it also depends on the size of the hydrometeors, the thickness of a given layer, the temperature of the layer, etc.This might be a bit over-simplified.

  We agree. This has probably been an oversight in condensing the abstract. This sentence now reads as:

  "Such radiative interactions rely, among other factors, on the ice content of MPC which is regulated by the availability of ice nucleating particles (INP)."

- Lines 34 and 35: For an earlier reference, also consider Creamean et al., 2018 (Creamean, J.M., R.M. Kirpes, K.A. Pratt, N.J. Spada, M. Maahn, G. de Boer, R.C. Schnell and S. China (2018):Marine and terrestrial influences on ice nucleating particles during continuous springtime measurements in an Arctic oilfield location, Chem. Phys., 18, 18023-18042, https://doi.org/10.5194/acp-18-18023-2018).

  We agree that this citation should have been considered given that their work were published earlier than the mentioned references. This reference has been added to the references of this sentence.

- Lines 35-37: Given that phase in high-latitude clouds and INP concentrations are both notoriously difficult to sample with satellite-based sensors, it seems that this sentence could use some more explanation (even if supported by a reference).

  We agree. We have added a sentence that explains that their results are reflected by ground-based remote sensing around the Arctic. The trade-off being that they are point based measurements. Thus a combination of satellite and ground-based cloud phase retrieval strengthen the argument.

  " Satellite observations show that the prevalence of MPC in the Arctic and Antarctic regions can be explained to a large degree by the presence of INP (Carlsen and David, 2022). Despite satellite remote sensing uncertainties, their results reflect those of ground-based remote sensing in the Arctic (Nomokonova et al., 2019)."

- Lines 168-169: "Here, we focus on an altitude of 400 to 600 meters." – presumably this is because that is the height of the Zeppelin Observatory, though it might be worth stating this directly.

  We agree, this was also noted by the other reviewer. We have added some clarifying sentences which read as follows:

  "Here, we analyze altitudes of 400 to 600 meters to reflect measurements taken at the Zeppelin Observatory altitude (475 meters asl)."

- Line 176: It isn't clear what is meant by "mixed ratio", and the term sounds a lot like "mixing ratio".I would consider rewording.

  We agree, all instances of "mixed ratio" were reworded as "ice-to-droplet ratio" to avoid confusion.

- Line 194: "within the mixed layer (as defined by the model/HYSPLIT output)":How confident are we in the fact that HYSPLIT and/or "the model" can do this correctly? Is there any evaluation of this for the Arctic? Please consider adding some language on uncertainties or potential challenges.

  We have previously evaluated that slightly increasing and decreasing the mixed layer height doesn't significantly influence the overall contribution of surface sources when using HYSPLIT. We have modified the text to better reflect the uncertainties that derive from the use of models. The new text reads as:

  "Data points along each and every back trajectory (i.e. endpoints) were selected only if they resided within the mixed layer (as defined by the model/HYSPLIT output). Previous work by Karlsson et al. (2021) showed

that increasing or decreasing the mixing layer height does not significantly affect the general contribution of surface types."

- Line 207: Consider rewording "the last part" to "finally" or similar to maintain some level of formality in the text.

  We thank the reviewer for this suggestion. This paragraph has been rewritten as:

  "First, we present an overview of fPBAP found in cloud residuals (Section 3.1 and 3.2). Second, we include an analysis of the annual cycle of all characterized CE (Sect. 3.3) and their source allocation (Sect. 3.4). Finally, we present a case study of a mixed-phase cloud event with the highest concentration of fPBAP (MPC, Sect. 3.5). Similar case studies for a liquid phase and an ice phase cloud are briefly presented and discussed in sections 1.1 and 1.2 of the SI."

- Line 274: "Close" should be "Closely".

  We have fixed this error in the revised manuscript.

- Line 303: I think that this is debatable. Shupe et al (https://psl.noaa.gov/people/matthew.shupe/publications/-Shupeetal.JAMC2011.pdf) shows that the fall is generally cloudier.

  Indeed. The clouds are more prevalent in fall months. However, for the purposes of this work we have bundled the months with sunlight into "summer" and those without sunlight into "winter" as Arctic seasons can often be misleading. We have slightly adjusted the text to improve clarity.

  "As can be seen, most CE were concentrated in summer, when sampling conditions were generally better. It is also documented that low-level clouds are also more present in the late summer (early fall) months (Illingworth et al., 2007; Taylor et al., 2019; Curry and Ebert, 1992; Maturilli and Ebell, 2018)."

- Line 306: There is an increased contribution of sea spray in winter months (over summer months)?Is this due to more storms? More explanation would be good.

  Yes, winter month increased wind speeds and higher prevalence of storms lead to higher sea salt/spray aerosol from both nascent sea spray and re-lifted snow. We have changed the text to better explain this relation:

  "This seasonality is due to increased wind speeds and prevalence of storms in winter, that generate nascent sea spray from ocean surfaces and lift sea salt rich snow from ice covered ocean (Adachi et al., 2022)."

- Line 319: "(1-5 and 9-12)" Is this supposed to be month numbers?If so, I would recommend just using the names of the months to avoid confusion.

  Yes, and we understand how these could be confusing we have decided to replace numbers with the month names. The new text reads as:

  "The ice-to-droplet ratio (Figures 3-C) shows that low-level clouds at Ny-Ålesund during the colder months (January-April and September-December) are mainly represented by ice clouds, whereas for July they are mainly represented by liquid droplets."

- Line 321: "as such were most-likely MPC"—no need for a hyphen between most and likely.Also, don't you have measurements to confirm this? If so, why say "most likely"?

  We were very careful on choosing our wording. Although, yes we do have remote sensing measurements that indicate the clouds to be MPC, they are not a direct measurement of the cloud phase. We decided to soften our caution in this sentence and have rewritten it as:

  "For June and August, most clouds were MPC, which reflects the findings of Mioche et al. (2015)."

- The paragraph starting on line 324 should be cleaned up, language-wise.I found it challenging to read and there were things like open parentheses, lots of commas, and unclear language. For example, the sentence structure "the altitude in which temperature reached -15 C sat above 2500 m" is unnecessarily complicated. Please re-read and re-word this section.

  We completely agree with the reviewer and have decided to restructure the paragraph. We hope that this change increases the clarity of this paragraph. The updated text reads as follows:

  "he ambient temperature at 475 meters asl reached values as low as -25 °C in winter and values as high as 10 °C in summer (Figure 3-D). In the months that transition from summer to winter, the air temperature was on average around 0 °C. For May through September, the ambient temperature reached -15 °C only at

altitudes higher than 2500 meters. This can be seen in Figure 3-E, derived by both daily radio soundings and continuous HATPRO vertical temperature profiling above the village of Ny-Ålesund. The cloud top height of low-level clouds was much higher in the beginning of the year, reaching its minima in summer, where it stayed below 2500 meters. Thus, the temperatures across the air columns in summer point to the requirement of high-temperature INP for ice formation to occur."

- Citation: https://doi.org/10.5194/egusphere-2023-2600-RC1

**2 Reviewer #2**

**2.1 General overview**

The paper comprises three case studies of an experiment to investigate the abundance of fluorescent primary biological aerosol particles, which may act as ice nuclei. The experiment was run over one and a half years, in the Svalbard archipelago, at an observatory atop a 500m mountain. Data was recorded from a Multiparameter Bioaerosol Spectrometer (MBS, to detect fPBAP), and a FIDAS 200S to measure size distribution, a Picarro L2130-i to measure isotopic ratio of oxygen and hydrogen. The description of the data and the experiment is extremely useful, and the data provided is novel, in that the size distribution and concentration of fPBAP which may act as ice nuclei is timely and of interest to cloud formation studies. However, the paper itself I found difficult to interpret at times, lacking in flow, and descriptions of some key elements. It is perhaps lengthier than necessary, and could be more concise without losing the key elements.

We thank the reviewer for their helpful comments. We would like to clarify that our paper included more than three case studies but also an overview and discussion on the annual cycle of fPBAP found in cloud residuals. This might have been missed due to the original structure of the manuscript. We therefore decided to reduce the case studies only to the mixed-phase cloud (the most important one) and also moved the overview/annual cycle to the front of the results and discussion section. The other two case studies of liquid and ice clouds are moved to the SI. We hope that the changes we have made to the overall structure of the paper increase the flow of the text at the same time as focusing on the most important aspects of our work.

**2.2 Major comments**

- Section 2.2 doesn't make it clear that the CVI is the component responsible for delivering samples to the MBS, TEM, and FIDAS 200S. The authors refer to the CVI as though it is measuring parameters of particles. This could be made clearer throughout.

We understand that this might not have been as thoughtfully laid out in the text as it could have been. For this, the methodology section of "Cloud Particle Sampling" has been worked on to properly reflect that the MBS and TEM measured cloud residuals remnant of captured cloud particles by the CVI while the FIDAS instrument sits on the roof sampling from its own inlet. The revised section reads as follows:

"Cloud droplets and ice crystals were collected using a ground-based counterflow virtual impactor (CVI) inlet (Brechtel Inc., USA, Model 1205). The CVI only collects particles above approx. $6\,\mu$m in aerodynamic diameter, representing aerosol particles that have been activated into cloud droplets or ice crystals. It does so by accelerating the cloud onto the CVI tip that is installed within a wind tunnel. Within the CVI tip a counterflow is targeted against the sample flow, where only larger particles have enough inertia to penetrate through the virtual impaction plate. A more technical description of the CVI is given in Noone et al. (1988) and Shingler et al. (2012), whereas a detailed characterization of the ground-based CVI present at the Zeppelin Observatory, together with the applied corrections, is given in Karlsson et al. (2021). After the cloud droplets and ice crystal penetrate through virtual impaction plate, they are dried in the counterflow air. The leftover nuclei are called cloud residuals, which are then sampled by the aerosol instrumentation downstream of the CVI. The measured cloud residual concentration after correcting for an enrichment factor (9.8 for this work) must be multiplied by a factor of 2, accounting for a mean droplet sampling efficiency of around $45\,\%$. This factor was determined by comparing the coarse-mode cloud residual particle concentration ($> 0.8\,\mu$m) measured by the MBS during CVI operation with the corresponding ambient (total) coarse-mode particle concentration measured by a FIDAS 200S (Palas GmbH, Germany) sampling from its own inlet located on the terrace of the Zeppelin Observatory (see Figure S1 in the SI). This value is comparable to the CVI sampling efficiency of $46\,\%$ previously determined by Karlsson et al. (2021)."

- The case studies are all short-duration (days) and spaced by months. It would be useful to know the reason for choosing these CE specifically, and also to contextualise them with some long-term data. Including outside of CE.

  The case studies were chosen by the highest presence of PBAP per type cloud. A new text was added to the first section of the results to better reflect the reasoning behind our case events. The new text reads as follows:

  "Finally, we present a case study of a mixed-phase cloud event with the highest concentration of fPBAP (MPC, Sect. 3.5). Similar case studies for a liquid phase and an ice phase cloud are briefly presented and discussed in sections 1.1 and 1.2 of the SI."

  We have chosen to move two of these cloud cases (liquid droplet and ice clouds) to the supplement as examples of these types of clouds. In the main text we have expanded the discussion of the MPC event and have placed it as the last section to properly contextualize it. During the description of the annual data, further references were made to our previous work at Zeppelin which analyzed the ambient fPBAP population.

- The conclusions section describes the experiment, and provides a brief description of the results. There are few impactful conclusions drawn from the data.

  As a similar comment was also brought up by the first reviewer, we have thoroughly restructured and overhauled the conclusion to better reflect the results and outlook of this work. We have removed unnecessary descriptions and highlighted the results of our work. The revised conclusion section reads as follows:

  "Within this work, we showed that fluorescent primary biological aerosol particles (fPBAP) are found within cloud residuals and possibly contributed to the formation of low-level Arctic clouds. This was achieved, for the first time, by direct observations using a ground-based counterflow virtual impactor inlet combined with online and offline particle sampling techniques. This approach avoided indirect proof of the relevance of fPBAP on cloud properties, for example, when using correlations of INP with fPBAP concentrations as done previously (Pereira Freitas et al., 2023). fPBAP exhibited higher concentrations ($10^{-3}$–$10^{-2}$ $L^{-1}$) and contributions (0.1 to 1 in $10^3$ particles) to the coarse-mode cloud residuals in summer compared to winter ($10^{-4}$–$10^{-3}$ $L^{-1}$, and 1 in $10^4$–$10^5$ particles, respectively). In summer, water vapor isotope data linked clouds to regional sources. Thus, fPBAP most likely originated from the biosphere around Svalbard. The presence of fPBAP was associated with the prevalence of mixed phase clouds at the beginning and end of summer. Here, we present experimental and direct evidences that fPBAP contribute to ice formation in Arctic low-level clouds. However, cloud formation is a complex phenomena involving meteorology as well as interlinked cloud and aerosol microphysical and chemical processes. Thus, the degree to which fPBAP influence cloud glaciation in general would require further investigation both experimental (e.g. by quantitative assessment of the cloud phase using single particle cloud probes) but also using modelling approaches. Future work should also include filter sampling for genetic analysis to identify the biological material and origin, in addition to parallel sampling of bioaerosols within cloud residuals and interstitial aerosol to assess whether certain microorganisms are more likely to act as cloud condensation nuclei."

**2.3 Minor comments**

- Line 168, "we focus on" suggests optical focussing of the remote sensing devices. Is that as intended, or are you extracting data for this region?

  We are extracting data for this region, we have revised this sentence to more clearly reflect our methodology. The text now reads:

  "Here, we analyze altitudes of 400 to 600 meters to reflect measurements taken at the Zeppelin Observatory altitude (475 meters asl).."

- Line 213, missing unit on concentrations

  Thanks. We have now provided the unit of these concentrations. The new text reads as follows:

  "These fPBAP concentrations are in the range of typical high-temperature INP concentrations found in the Arctic ($10^{-4}$–$10^{-1}$ $L^{-1}$, at activation temperature $\approx$-15°C, Creamean et al. (2022); Sze et al. (2023); Pereira Freitas et al. (2023))."

- The MBS includes shape classification capabilities, which may help with discriminating cloud phase. It would be useful to explain why these weren't employed (i.e., in order to measure the nuclei, the sample was dried)

Given that the MBS was only measuring the dried cloud particles (and not the ambient cloud properties) this is unfortunately not possible. We have analyzed briefly the shape classification parameters and found no meaningful insights that would be worth discussing in this work. To add better clarity to the text, some changes in the text were made in the methodology section "Cloud Particle Sampling". The revised text reads as follows:

[revised manuscript text omitted]